# Integration of Digital Technologies in Corporate Social Responsibility (CSR) Activities: A Systematic Literature Review and Bibliometric Analysis

Atanas Atanasov [ID], Galina Chipriyanova [ID] and Radosveta Krasteva-Hristova *[ID]

Department of Accounting, Tsenov Academy of Economics, 5250 Svishtov, Bulgaria;
a.atanasov@uni-svishtov.bg (A.A.); g.chipriyanova@uni-svishtov.bg (G.C.)
* Correspondence: r.krasteva@uni-svishtov.bg

**Abstract:** Modern technologies require the need to analyze the opportunities for improving the integration of digital technologies in CSR activities in the context of added values between business and science in perspective, including the future digital society. The main goal of this article is to identify the current state of research on the integration of digital technologies in CSR activities in business, as well as to prepare recommendations for further research and practice. Additionally, the study aims to recognize the relationship and dependencies between CSR and digital technologies. A systematic literature review and bibliometric analysis of 129 scientific articles published between 2014 to 2023 was performed. The bibliometric analysis was organized in two directions: descriptive and performance analysis, through which we can study the contribution of the analyzed objects to the given scientific area, and science mapping, which studies the relationships among them. The results indicate that companies more frequently use artificial intelligence, blockchain, the Internet of Things and other technologies to increase the efficiency and impact of their CSR activities. In addition, this research reveals the basis of bringing forward the new trends for future publications, which shall upgrade and enrich the theory and practice.

**Keywords:** corporate social responsibility; digital technologies; systematic literature review; bibliometric analysis

## 1. Introduction

In today's national and international socio-economic conditions, the issues related to engaging those who manage businesses, the public asector, and non-governmental organizations in social causes are becoming increasingly relevant. To support this claim, we can point out numerous managerial solutions and activities oriented towards the specific interests (including those related to the environment) of customers, suppliers, personnel, and the society as a whole within the scope of the economic and non-economic activity they realize. The outlined solutions and activities have acquired a public character via the collective notion "corporate social responsibility" (CSR), which has been the object of a number of theoretical and practical analyses and discussions (Krysovatyy et al. 2022; Lamptey et al. 2023; Shahadat et al. 2023; Singh and Hong 2023). CSR is a concept whereby companies voluntarily consider the social and environmental aspects of their activities and, in this context, make commitments to society and the environment that go beyond their primary goal of profit-making (Carroll 1999). It is concerned with treating the stakeholders of the firm ethically or in a responsible manner. Some companies consider CSR as corporate philanthropy, while others as a new corporate strategic framework (Hopkins 2004). CSR now includes social, environmental, and governmental goals that are connected with companies' missions. The level of corporate social responsibility can be perceived differently for individual enterprises (Dudek et al. 2023). From previous literature, it was found that popular digital tools are successfully used in finance, management, industry, and education

(Muchiri et al. 2022; Pangalos 2023; Paris et al. 2023; Richert and Dudek 2023; Sheel and Nath 2019). Digitalization is powerful because it not only allows for automation but also tracks and stores information and data about tasks and activities, thereby creating a record that can be analyzed and that provides opportunities to improve processes, work organization, and predictions about future events (Ciarli et al. 2021). Digital technologies change all economic and social activities and result in the creation of innovative ideas, services, and business opportunities. Furthermore, the expansion of their consistent and long-term implementation combined with their integration in CSR activities is an undisputed testimonial of the existence of foresight and vision in management.

In the conditions of a permanently digitalizing environment, corporations increasingly use digital technologies. Their goal is to improve their CSR activities by constantly perfecting the measurement of their efficiency, communication with stakeholders, and the competitiveness on the market. This research presents a theoretical and applied study, which combines the objective capacity of a bibliometric analysis and the subjective views of the authors regarding the topic and the scope of the researched materials, integrated in scientific conclusions, trends, and perspectives of future research.

The main goal of the research is to identify the current state of scientific research in the field of integrating digital technologies in activities related to CSR in business, as well as to propose recommendations for further research and practices. Additionally, the study aims to recognize the relationship and dependencies between Corporate Social Responsibility and digital technologies.

The interpretation of the relationship between CSR and digital technologies is of crucial importance, as in today's digitized world, companies are becoming increasingly dependent on technologies to communicate and to function effectively and efficiently. The connection between these two concepts can be expressed in the following aspects:

- Ethical and Responsible Handling of Data—digital technologies allow companies to collect, process, and analyze large volumes of personal data from consumers. In the context of CSR, companies must demonstrate ethical conduct and responsible behavior in collecting and managing these data. This includes respecting personal privacy, providing clear information about the purposes of data processing, and ensuring the protection of personal data from misuse and breaches (Quach et al. 2022);
- Transparency and Accountability in Technology Utilization—digital technologies have the potential to facilitate the establishment of transparent and accountable business practices. Companies that aspire to be socially responsible must ensure clear and understandable information regarding how they employ their technologies and the impact they have on society and the environment. Publicly disclosing information about corporate digital practices is a crucial element in the endeavor to enhance trust in the company (Bekele 2023);
- Opportunities for Innovation and Social Challenges—digital technologies offer unique opportunities for companies to innovate and create new products and services that can address social challenges, contributing to sustainability and improving quality of life (Dionisio et al. 2023);
- Reducing Digital Divide—digital technologies present countless opportunities for innovation and growth. Companies committed to CSR play a crucial role in reducing the digital divide by striving to ensure access and equal opportunities for digital technologies across all layers of society (Basu 2017);
- Cybersecurity and Customer Protection—cybersecurity becomes an increasingly vital aspect of corporate responsibility in the context of the increasing digitization of business and society. Companies must invest in cybersecurity and protect their customers' data to prevent abuses and breaches of personal information (Serabian 2015).

Achieving the research goal is related to the successive completion of several stages in which a specific methodology and technical set of tools are applied (Figure 1).

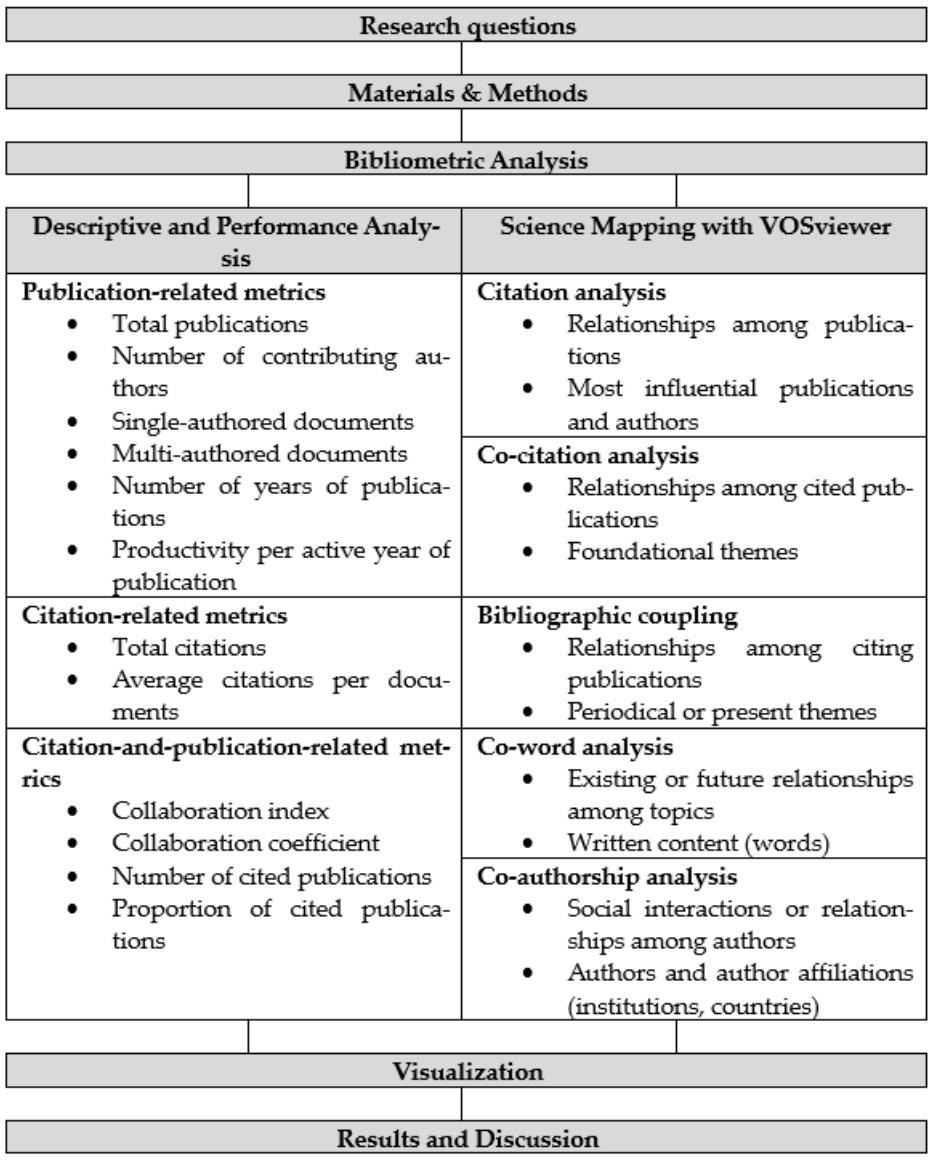

**Figure 1.** Research Flowchart. Adapted from (Donthu et al. 2021).

The research design, to a large extent, is constructed in compliance with the set goals and objectives. A key stage in the research process is the definition of research questions, which shall facilitate the identification and analysis of the existing literature, the research on the interrelations among various topics, concepts, and authors, to outline the achievements, gaps, and future guidelines for scientific research in the field of integrating digital technologies in CSR activities.

To achieve the goal of the article, we have set the following research questions (objectives):

- RQ1: What are the most influential publications, authors, and institutions that deal with the research on digital technologies in CSR activities?
- RQ2: How has the literature in the field of integrating digital technologies in CSR developed historically and how has it spread geographically?
- RQ3: What is the scope of research, and what are the key trends for future scientific research in the field of integrating digital technologies in CSR activities?
- RQ4: Is there an integration of digital technology into the CSR activities?

The main research problem addressed in the present paper is that CSR activities are thoroughly theoretically and scientifically grounded and digital technologies have the

capacity to grow and successfully support CSR activities—a practice which, without doubt, contributes to their accelerated development and implementation, with a horizon towards corporate social responsibility.

## 2. Materials and Methods

The research materials have been generated using Scopus—the largest abstract and citation database, which as of this moment, contains over 87 million documents back to the year 1788, over 1.8 billion cited references back to the year 1970, over 17 million author profiles, and over 7000 publishers, including those in the field of social sciences. Despite the comprehensive coverage with open-access content, Scopus provides a wide range of metrics and tools to evaluate the research impact and collaborating author profiles and create collaboration networks. At this stage of the scientific research, we have determined a series of key words and terms and a search field in the database, a selection of additional search criteria; we have also generated a sample containing data about the research. The protocol that we have applied is Preferred Reporting Items for Systematic Reviews and Meta-Analyses (PRISMA) (Page et al. 2021).

The applied research methods—Systematic Literature Review (SLR) and Bibliometric Analysis—have been selected in compliance with the set goal. They are used to generate quantitative data regarding publications, sources, authors, organizations, countries, key words, topics, and trends (Monteiro et al. 2021; Van Eck and Waltman 2010; Warren et al. 2019; Yu et al. 2020). This methodology is considered appropriate for collecting complex academic data and proof for the state and trends of integrating digital technologies in corporate social responsibility activities in enterprises.

We, as authors, have selected some of the specialized literature and performed a Systematic Literature Review (SLR) because the process of its performance is "structured, reproducible, transparent, and iterative in nature" (Abdullah and Naved Khan 2021) and provides an objective basis for eliminating irrelevant research. We have performed a review of the abstracts and full texts of the materials—the object of this scientific research. There are notions that traditional literature reviews suffer from the subjectivity of the authors (Tranfield et al. 2003), which is an additional reason SLR should be used. As a result, the performed categorization and summary of information with a clearly defined goal, research questions, search methods, and inclusion and exclusion criteria in the search allow the scientific articles to be given a high grade.

The collected data about the published scientific research and citations is subjected to a Bibliometric Analysis to determine its impact. The aim of the bibliometric analysis is to "summarize large quantities of bibliometric data to present the state of the intellectual structure and emerging trends of a research topic or field" (Donthu et al. 2021). The objects of analysis can be scientific publications (articles, papers, books), sources (scientific journals, conference proceedings), characteristics of the researched area key words and terms, authors, institutions (universities, departments, scientific and research groups and teams, business organizations, NGOs), countries, and regions.

The bibliometric analysis is performed in two directions: Descriptive and Performance Analysis, through which we can study the contribution of the analyzed objects to the given scientific area, and Science mapping, which studies the relationships among them.

Science mapping is a set of methods and techniques through which we can develop science maps to reveal the structural and dynamic aspects of scientific knowledge (Börner et al. 2005; Cobo et al. 2011; Morris and Van Der Veer Martens 2009; Noyons et al. 1999; Petrovich 2021). They aim at showing how areas, disciplines, journals, scholars, publications, and key words are related among each other.

Bibliographic databases, including Scopus, are the main sources of data for science mapping. These enormous multidisciplinary databases collect metadata of academic publications (authors, title, abstract, key words, affiliation of the authors, publication year, country), along with their citations.

The creation of science maps is based on networks via procedures called "network extraction". Networks are structures compiled by nodes and links, which are generated by a set of bibliographic records. The same set of bibliographic records can generate various networks depending on the type of nodes and links on which we focus. The nodes are the units of analysis of the end map, whereas the links are the types of the shown link. The networks can capture the scientific structure at various levels and from different points of view (Petrovich 2021).

The main techniques of scientific mapping, the goals, the relationships (links) and the units of analysis (nodes), as well as the necessary metadata, are shown in Table 1.

**Table 1.** Main techniques for science mapping *.

| Technique | Goal | Link | Node | Metadata |
|---|---|---|---|---|
| Citation analysis | To identify the most influential authors, publications, and sources in the research field | Relationships among publications | Documents | Author name Citations Title Journals References |
| Co-authorship analysis | To identify the most influential authors, institutions, or countries in the research field | Relationships among authors and their affiliations | Authors Affiliations | Author Affiliation (institution and country) |
| Bibliographic coupling | To understand the themes and the scope in a research field | Relationships among citing publications | Documents | Author name Title Journals DOI References |
| Co-citation analysis | To understand the development of the foundational themes in a research field | Relationships among cited publications | Documents | References |
| Co-word analysis | To explore the existing or future topics in a research field | Relationships among topics | Words | Title Abstract Author key words Index key words Full text |

* Adapted from (Donthu et al. 2021).

The tool that we have selected for visualizing science maps in this research is VOSviewer (version 1.6.19). The mapping profiles comply to the above research questions, to the theoretical models (cf Table 1), and the software capacity.

## 3. Results

### 3.1. Data Collection

- Defining key words, terms, and the field of search in the database. Defining key words and terms for searching the Scopus database is the result of a panel discussion among the research authors and specialists in the field of information technologies and the implementation of CSR in business. Searching and extracting during the data collection phase involve the use of different search techniques, such as Booleans, use of quotation marks and parenthesis, and truncations. The selected key words run in two directions: regarding the applicable digital technologies ("digit*" OR "digital technolog*"), and,

regarding the field of CSR ("CSR" OR "corporate social responsib*"). Appropriate Boolean operators OR and AND have been used. The query in the database based on these key words was conducted on 9 June 2023 in a specific field (Article title, Abstract, Key word) and the generated result was 899 documents.

- Selection of additional search criteria. The selected PRISMA protocol requires defining additional search criteria that limit the sample. Those criteria are selected by the authors who have used the search filters suggested in the Scopus database. The selected criteria options limit the sample to the following:

1. Open Access—All open access;
2. Year—we have selected a period of published articles between 2014 and 2023;
3. Subject area—we have selected the areas Business, Management and Accounting, Social Sciences; Environmental Science; Computer Science; and Economics, Econometrics, and Finance;
4. Document Type—we are only interested in articles/Article;
5. Publication Stage—only completed articles with the option Final are selected;
6. Source Type—we have selected articles published in journals Journal; and
7. Language—the research encompasses articles in English only with the option English.

The remaining filters Author name, Source title, Key word, Affiliation, Funding Sponsor, and Country/territory have not been used.

The query in the database after applying those additional criteria ranks the data and forms a sample of 129 documents.

- Cleaning the data. A very important part of the process of data processing is its cleaning. It is related to eliminating some of the records. The PRISMA protocol requires that the research materials (in this case 129 scientific publications) be checked for duplicates to eliminate them. The check has shown that there are no duplicates. Another reason for cleaning the data is the review and assessment of the articles to determine their relevance and eligibility. The titles and abstracts of the selected documents in the sample have been reviewed by the authors. After an in-depth discussion, all articles in the sample have been accepted as relevant to the scientific research.
- Data storage. The extracted data have been exported to a .csv file and stored for the purposes of their analysis. In addition, we have stored the full text copies of the articles in the sample. The query has been stored in the Scopus database.

*3.2. Performing Descriptive and Performance Analysis*

The materials, which are the object of this research, are a collection of 129 scientific articles written by a total of 443 authors; 14 out of 129 articles were written by one author. Among those 14 articles, however, two were written by the same author—M. A. Camilleri (in 2018 and in 2019). Another characteristic is that B. Lopez has two articles—one as the only author from 2022 and one as a co-author from 2023. We believe that these circumstances are insignificant regarding the results and are not taken into consideration further in the analysis. The time frame of the research is 10 consecutive years between 2014 and 2023. The average number of published articles per year is almost 13 (more precisely—12.9 articles).

Regarding citations, we have used ready information, which has been generated as a result of the query in Scopus. Not all 129 in the sample are cited in other scientific publications. The number of the cited articles is 98, which is 75.97% of the sample volume. The total number of citations of these 98 articles in other scientific publications is 1995. The average number of citations of these articles is 20.36 times.

The focus of this analysis also comprises citation-and-publication-related metrics: collaboration index, collaboration coefficient, number of cited publications, and proportion of cited publications.

The Collaboration Index (*CI*) of authors shows the average number of authors per one article. The *CI* distinguishes the levels of authorship and is easy to calculate, but there are no boundaries (for instance between 0 and 1), which makes it difficult to interpret.

The data show that the *CI* is 3.73 or that the research teams have comprised 3 or 4 researchers on average. The collaboration index is calculated as the ratio of Total Authors of Multi-Authored Articles (*TAMAAs*) and Total Multi-Authored Articles (*TMAAs*) (1) (2) (Elango and Rajendran 2012):

$$CI = \frac{TAMAAs}{TMAAs} \tag{1}$$

$$CI = \frac{443 - 14}{129 - 14} = \frac{429}{115} \approx 3.73 \tag{2}$$

Another thing to take into consideration is the Collaborative Coefficient (*CC*), which is used to measure the degree of collaboration among the authors who have contributed to the literature regarding the integration of digital technologies in CSR activities. This index may vary in the interval between 0 and 1. The higher than 0.5 it is, the better the collaboration ratio among authors. When it is close to 0, this means that there is poor collaboration among authors. In this particular case, the *CC* has a value of 0.62. To calculate it, we have used the following formulas (3) and (4) (Ajiferuke et al. 1988):

$$CC = 1 - \frac{\sum_{j=1}^{k}\left(\frac{1}{j} \cdot F_j\right)}{N} \tag{3}$$

$$CC = 1 - \left(\left(\frac{1}{1}\cdot 14 + \frac{1}{2}\cdot 30 + \frac{1}{3}\cdot 21 + \frac{1}{4}\cdot 19 + \frac{1}{5}\cdot 22 + \frac{1}{6}\cdot 14 + \frac{1}{7}\cdot 5 + \frac{1}{8}\cdot 3 + \frac{1}{9}\cdot 1\right) : 129\right) = 0.62 \tag{4}$$

In this formula, *j* is the number of authors of one article, $F_j$ is the number of articles with *j* authors, *N* is the total number of published scientific articles, and *k* is the largest number of authors of one article in the sample, as shown in Table 2. It should be noted that the collaboration coefficient can be calculated for a specific author, in which case, the scientific publications to which he/she has contributed will be the volume of the researched sample of documents.

**Table 2.** Distribution of the number of articles based on the number of authors by years *.

| Years | Articles per Year | Articles per Author | | | | | | | | |
|---|---|---|---|---|---|---|---|---|---|---|
| | | *j* = 1 | *j* = 2 | *j* = 3 | *j* = 4 | *j* = 5 | *j* = 6 | *j* = 7 | *j* = 8 | *j* = 9 = *k* |
| 2014 | 1 | 0 | 0 | 0 | 0 | 0 | 0 | 0 | 1 | 0 |
| 2015 | 2 | 0 | 1 | 1 | 0 | 0 | 0 | 0 | 0 | 0 |
| 2016 | 4 | 1 | 0 | 0 | 1 | 1 | 0 | 0 | 1 | 0 |
| 2017 | 7 | 1 | 1 | 1 | 0 | 0 | 3 | 1 | 0 | 0 |
| 2018 | 3 | 1 | 2 | 0 | 0 | 0 | 0 | 0 | 0 | 0 |
| 2019 | 10 | 1 | 3 | 1 | 1 | 2 | 1 | 1 | 0 | 0 |
| 2020 | 13 | 1 | 4 | 3 | 2 | 2 | 1 | 0 | 0 | 0 |
| 2021 | 29 | 2 | 8 | 3 | 5 | 4 | 4 | 2 | 1 | 0 |
| 2022 | 34 | 5 | 7 | 7 | 4 | 8 | 2 | 0 | 0 | 1 |
| 2023 | 26 | 2 | 4 | 5 | 6 | 5 | 3 | 1 | 0 | 0 |
| Total | *N* = 129 | $F_1$ = 14 | $F_2$ = 30 | $F_3$ = 21 | $F_4$ = 19 | $F_5$ = 22 | $F_6$ = 14 | $F_7$ = 5 | $F_8$ = 3 | $F_9$ = 1 |

* Prepared by authors.

The results of the Descriptive and Performance Analysis of the data have been summarized in Table 3.

**Table 3.** Performance analysis of retrieved articles in the sample *.

| Publication-related metrics | Results |
|---|---|
| Total publications (TPs) | 129 |
| Number of contributing authors (NCA) | 443 |
| Single-authored documents | 14 |
| Multi-authored documents | 115 |
| Number of years of publications (2014–2023) (Y) | 10 |
| Productivity per active year of publication (PAY = TP/Y) | 12.9 |
| **Citation-related metrics** | **Results** |
| Total citations of 98 documents (in Scopus) (TC) | 1995 |
| Average citations per document (AC = TC/TP) | 20.36 |
| **Citation-and-publication-related metrics** | **Results** |
| Collaboration index (CI) | 3.73 |
| Collaboration coefficient (CC) | 0.62 |
| Number of cited publications (NCP) | 98 |
| Proportion of cited publications (PCP = NCP/TP) | 0.76 |

\* Adapted from (Donthu et al. 2021).

The data can also show the dynamics of publications in the researched area (Figure 2). The visualization allows a clearly outlined exponential development of the researched period to be found, which shows an increased interest in studying the problems in the field of implementing digital technologies in CSR activities by enterprises. However, when we are considering the growth of articles, we acknowledge that growth may be exponential at this point, but it could be following a sigmoid curve (Nikulin et al. 2011). In this case, the growth of articles looks sigmoid, but we assume that this is not a sigmoid, but an exponential, curve. The reason for this is that the last period, which consists of 6 months, does not correspond to previous years (12 months).

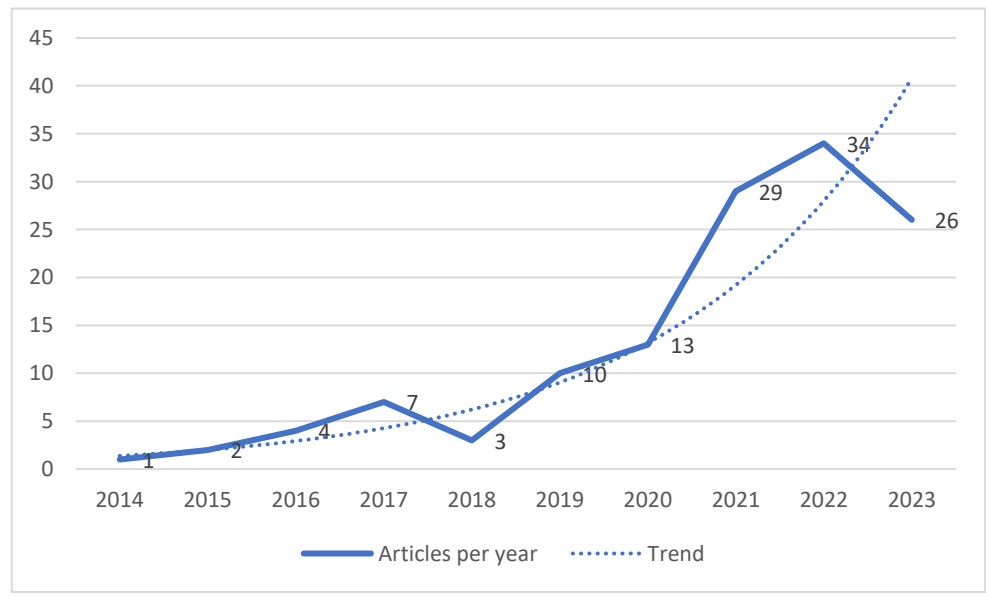

**Figure 2.** Articles published per year. Prepared by authors.

The sample comprises publications of which authors are affiliated with 160 scientific institutions and organizations, published by 51 countries in 87 respected peer-reviewed journals in the fields of business, management, economics, environment, and computer sciences. There is no information on the origin of three of the researched articles; therefore,

the analysis related to the geographic spread of the researched topic encompasses the remaining 126 articles.

The distribution of the number of published articles (total publications, TPs) by countries/regions is illustrated in Figure 3. The top three of countries with the most publications in the researched field are China with 25 published articles, followed by the United Kingdom with 22 and the United States with 15.

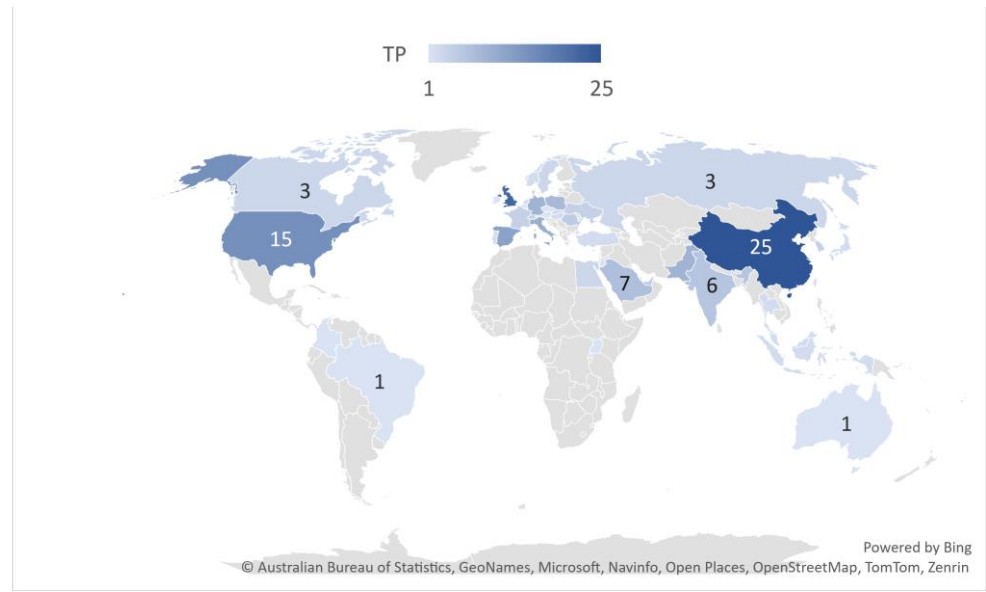

**Figure 3.** Articles published by country/region. Prepared by authors.

The quantitative information, which also refers to the number of citations of documents in the sample in the countries/regions where they are published, provides an opportunity to rank them. The first 10 records based on this criterion are shown in Table 4.

**Table 4.** Top 10 countries/regions with the highest number of document citations from the sample *.

| Rank | Country/Region | Total Citations | Total Cited Documents |
|------|----------------|-----------------|------------------------|
| 1 | United Kingdom | 675 | 22 |
| 2 | United States | 573 | 16 |
| 3 | France | 483 | 4 |
| 4 | China | 466 | 25 |
| 5 | Denmark | 397 | 2 |
| 6 | Switzerland | 311 | 4 |
| 7 | Spain | 221 | 12 |
| 8 | Italy | 186 | 10 |
| 9 | Pakistan | 152 | 9 |
| 10 | Germany | 108 | 9 |

* Prepared by authors.

### 3.3. Performing Science Mapping with VOSviewer

3.3.1. Citation Analysis

Citation analysis is a means of measuring the interest in a publication compared to the total number of its citations in other scientific publications. We acknowledge that articles can be cited for positive or negative reasons, and citation counts serve as an indicator that researchers are discussing a particular issue.

VOSviewer allows for the profiling of citation analysis in the following directions: documents, sources, author, organizations, and countries.

- Citation analysis by documents. The citation analysis when the analysis criterion documents is selected presents the articles based on their being cited in other documents

across the whole Scopus database. At this stage, VOSviewer allows only those articles for which a minimum number of citations of a document is selected by the user to be selected. The authors have selected 3 as the number of citations. This criterion is met by 72 documents in the sample, which is approximately 56% of the total number of 129 articles. At the next stage, the software brings forward the list of documents identified with the names of authors and the year of their publication, which contains two types of information: (1) the number of citations of those 72 documents in Scopus; and (2) the number of mutual citations of those 72 documents in the remaining documents in the sample, which form links among them.

- The science map shows the most influential publications and authors (the largest circles), but also that the sample indicates an extremely low level of citation links of the articles. Those that have cited each other are colored because they form clusters. The largest clusters are in red (three items) and in green (three items).
- In a search of the most influential publications and authors, we have performed a ranking of the articles in a sample based on the total number of citations in Scopus and the formed clusters compared to the number of articles in the sample linked by citing. We have performed normalization based on the number of citations, which is an option in VOSviewer. The results of this in-depth analysis are shown in Table 5, whereas the science map is shown in Figure 4. The science map shows that two clusters are positioned in it; this is illustrated in Figures 5 and 6.

**Table 5.** Citation analysis by documents (results) *.

| Documents | Citation in Scopus | Citation in the Sample | VOSviewer Cluster |
|---|---|---|---|
| (Scherer et al. 2016) | 241 | 0 | No |
| (Iglesias et al. 2020) | 156 | 0 | No |
| (Okazaki et al. 2020) | 63 | 2 | Red |
| (Jayaram et al. 2015) | 63 | 0 | No |
| (Etter et al. 2019) | 52 | 0 | No |
| (Jernigan and Ross 2020) | 46 | 0 | No |
| (Siano et al. 2016) | 44 | 0 | No |
| (Dreyer et al. 2017) | 39 | 0 | No |
| (Gupta et al. 2021) | 34 | 0 | No |
| (Ahmad et al. 2021b) | 34 | 0 | No |
| (Cheng et al. 2021) | 34 | 0 | No |
| (Camilleri 2019) | 34 | 0 | No |
| (Orbik and Zozul'aková 2019) | 15 | 2 | Green |
| (Khattak and Yousaf 2022) | 13 | 1 | Red |
| (Liu and Jung 2021) | 6 | 1 | Green |
| (López 2022) | 5 | 1 | Red |
| (Aitken et al. 2021) | 5 | 1 | Green |
| Total | 884 | 8 | NaN |

* Prepared by authors.

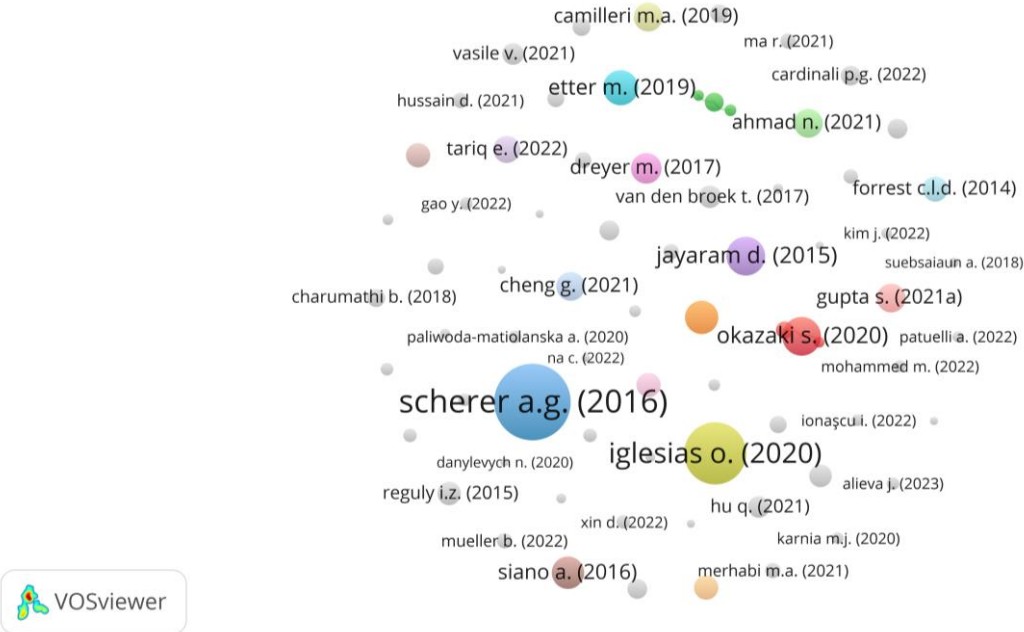

**Figure 4.** Scientific map of citation analysis by documents. (Ahmad et al. 2021b; Alieva and Powell 2023; Camilleri 2019; Cardinali and Giovanni 2022; Charumathi and Gaddam 2018; Cheng et al. 2021; Danylevych and Poplavska 2020; Dreyer et al. 2017; Etter et al. 2019; Forrest et al. 2014; Gao and Jin 2022; Gupta et al. 2021; Hu et al. 2021; Iglesias et al. 2020; Ionaşcu et al. 2022; Jayaram et al. 2015; Ma et al. 2021; Merhabi et al. 2021; Mohammed et al. 2022; Mueller 2022; Na et al. 2022; Okazaki et al. 2020; Paliwoda-Matiolanska et al. 2020; Patuelli et al. 2022; Reguly and Giles 2015; Scherer et al. 2016; Siano et al. 2016; Suebsaiaun and Pimolsathean 2018; Tariq et al. 2022; van den Broek et al. 2017; Vasile et al. 2021; Xin et al. 2022).

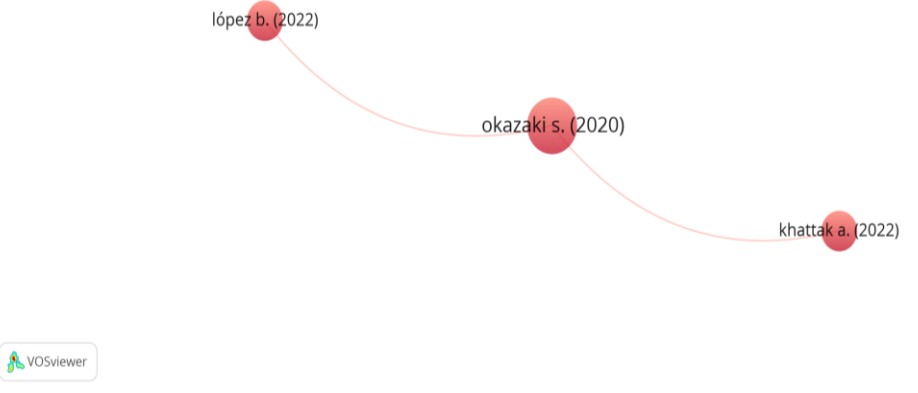

**Figure 5.** Scientific map of citation analysis by documents (red cluster). (Khattak and Yousaf 2022; López 2022; Okazaki et al. 2020).

- Citation analysis by authors. The citation analysis by authors reveals information about the author with the most articles and the most citations. The objects of the analysis are the articles and citations of all 443 authors of the 129 researched articles. The authors have been selected on the basis of a minimum number of articles and minimum number of citations (in Scopus). In this particular case, for both criteria we have selected a value of 1. This restriction returns a list of 289 authors, who, against the total number of 443, are 65.2%. In the list, the authors are ranked on the basis of the number of authored articles, the total number of citations of those articles in Scopus and the number of citations of one article in one journal in an article in another journal out of the totality of articles by 289 authors. Based on the last criterion, the values are between 8 and 0, which means that the authors of articles in the sample are cited

between 8 and 0 times by other authors in the same sample. Some of the authors have a substantial number of citations in the Scopus database (for instance, Ahmad, N.); however, they have been cited by none of the authors in this sample. The science map shows several clusters, which show in reality several multi-authored publications. The science map is visualized with VOSviewer and is shown in Figure 7.

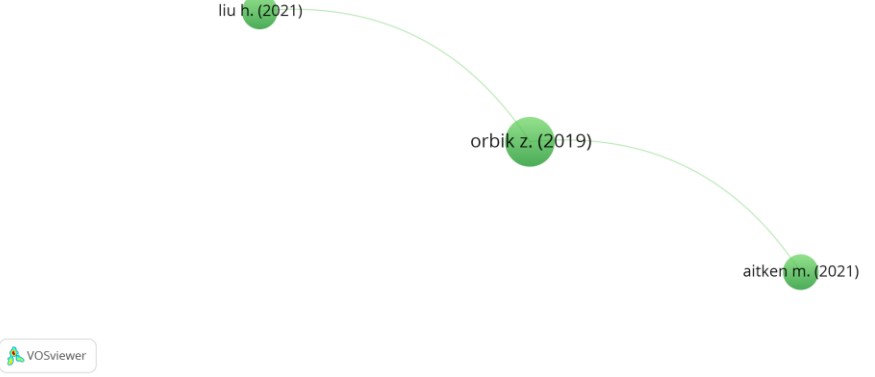

**Figure 6.** Scientific map of citation analysis by documents (green cluster). (Aitken et al. 2021; Liu and Jung 2021; Orbik and Zozul'aková 2019).

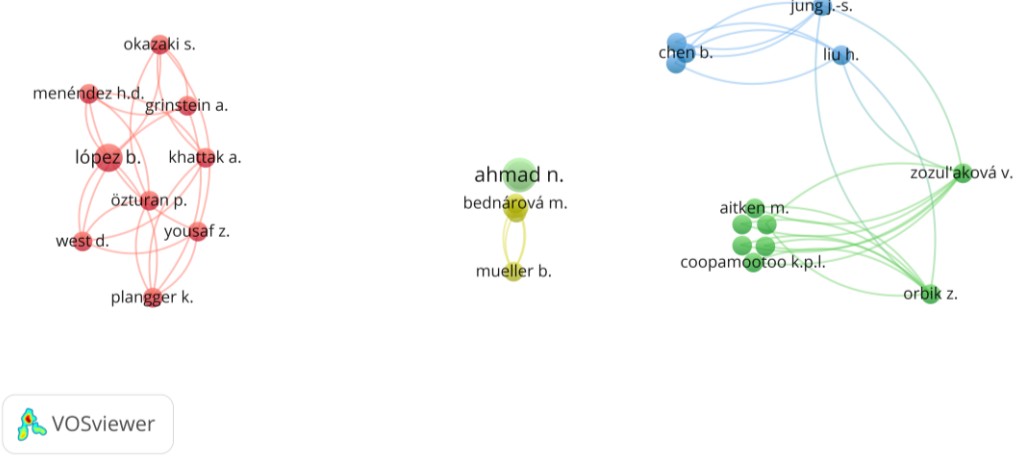

**Figure 7.** Scientific map of citation analysis by authors.

- Citation analysis by sources. The citation analysis can also generate data about the most influential scientific journals in which articles are published. The object of the analysis is those 87 sources in which all 129 documents have been published. The ranking of the journals is performed simultaneously based on the number of articles in the sample in a specific journal, the total number of citations of those articles in all remaining documents across the whole Scopus database, and the number of citations of one article in one journal by another journal out of the totality of 87 items. At this stage, VOSviewer allows only those articles for which the minimum number of documents of a source and minimum number of citations of a source are selected by the user to be selected. In this research, the authors have selected, for both options, a value of 2. The restrictions are met by **nine sources** out of 87, i.e., 10.3% of the total sources. At the next stage, the software brings forward a list of the journals, which contains information in three directions: (1) number of articles in each of the nine journals; (2) number of citations in Scopus of those articles that have been published in the selected journals; and (3) the mutual citations of articles within those nine journals in the sample, which form among each other, relationships of specific strength.
- The science map shows the most influential publications (the largest circles). It is important to note that there are only two journals in the sample that have a relationship

between them, but the visualization does not allow this relationship to be revealed. These are the journals Sustainability (Switzerland) and Frontiers in Environmental Science, which have mutually cited one article per journal. There is no level of connectivity via citations among the remaining journals in the sample.

- To reveal the most influential journals in which the articles in the sample are published, we have performed a ranking of the nine journals based on the number of published articles in each of them, the total number of citations of the articles in Scopus, and the clusters with the number linked by citing articles in the sample. The results of this in-depth analysis are shown in Table 6, whereas the science map is shown in Figure 8.

**Table 6.** Citation analysis by sources (results) *.

| Source | Published Articles | Citations in Scopus | Citations in the Sample | VOSviewer Cluster |
|---|---|---|---|---|
| Sustainability (Switzerland) | 25 | 262 | 1 | Red |
| Journal of Business Ethics | 4 | 232 | 0 | No |
| Journal of Theoretical and Applied Electronic Commerce Research | 2 | 68 | 0 | No |
| Information (Switzerland) | 2 | 32 | 0 | No |
| International Journal of Environmental Research and Public Health | 6 | 26 | 0 | No |
| IEEE Access | 2 | 16 | 0 | No |
| Frontiers in Environmental Science | 2 | 3 | 1 | Red |
| International Journal of Innovative Technology and Exploring Engineering | 2 | 3 | 0 | No |
| Water (Switzerland) | 2 | 2 | 0 | No |
| Total | 47 | 644 | 2 | NaN |

* Prepared by authors.

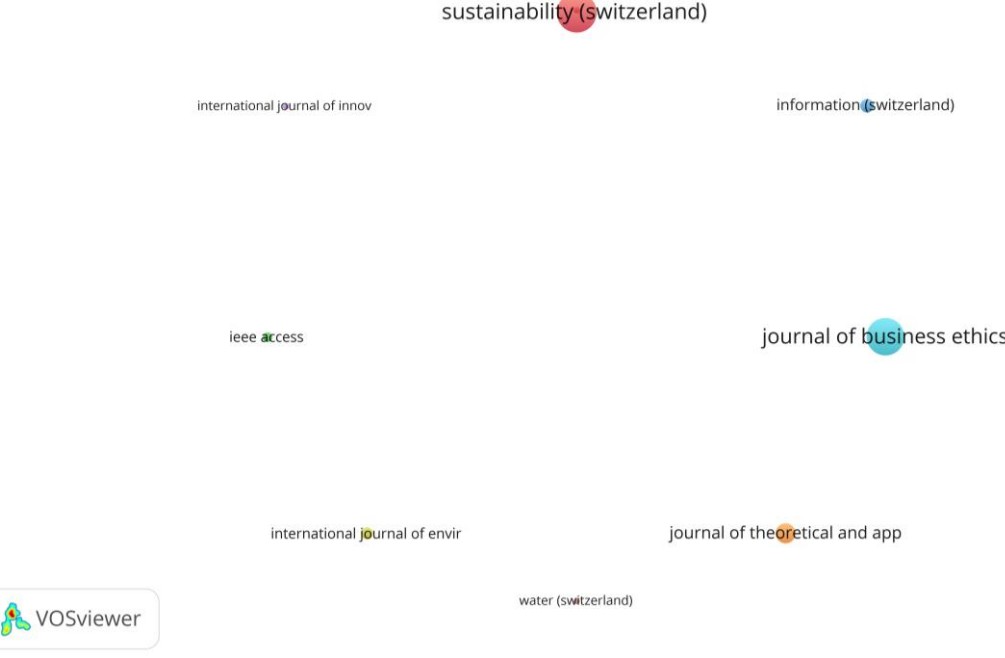

**Figure 8.** Scientific map of citation analysis by sources.

### 3.3.2. Co-Authorship Analysis

VOSviewer is a tool that can be used to analyze co-authorship in three directions—by authors, by institutions, and by countries. For the purposes of this publication, we have performed a co-authorship analysis only by countries. It refers to analyzing and visualizing collaboration models among authors from different countries on the basis of their co-authorship relationships.

The object of co-authorship analysis is, in fact, 51 countries in which the articles in the researched sample have been published. This type of analysis aims at outlining the international collaboration network within the researched topic and to understand the degree of collaboration among authors from different countries. At this stage, VOSviewer allows only those articles for which a minimum number of documents of a country and minimum number of citations (in Scopus) of a country are selected by the user to be selected, and in this case, their values are respectively 2 and 5. The restrictions are met by 30 countries. The ranking of countries is performed simultaneously based on the number of articles in the sample that have been published by a specific country, the total number of citations of those articles in Scopus, and the number of citations of one article by one country in an article by another country out of the selected 30 countries.

The science map shows in which countries the authors collaborate as co-authors of publications and creates a co-authorship network. Thus, we also learn about the geographic distribution of the publications in the field of digitalization and CSR.

VOSviewer identifies clusters or groups of authors from the same or different countries who frequently collaborate. The visualization can help identify the major centers of international collaboration and understand the whole network structure. The results of this in-depth analysis are shown in Table 7, whereas the science map is shown in Figure 9.

**Table 7.** The co-authorship analysis by countries (results) *.

| Country | Published Articles | Citations in Scopus | Citations in the Sample | VOSviewer Cluster |
|---|---|---|---|---|
| United Kingdom | 19 | 617 | 10 | Yellow |
| Pakistan | 9 | 152 | 9 | Green |
| Spain | 12 | 221 | 6 | Blue |
| Saudi Arabia | 7 | 88 | 6 | Purple |
| Germany | 9 | 108 | 6 | Red |
| Total | 56 | 1186 | 37 | NaN |

* Prepared by authors.

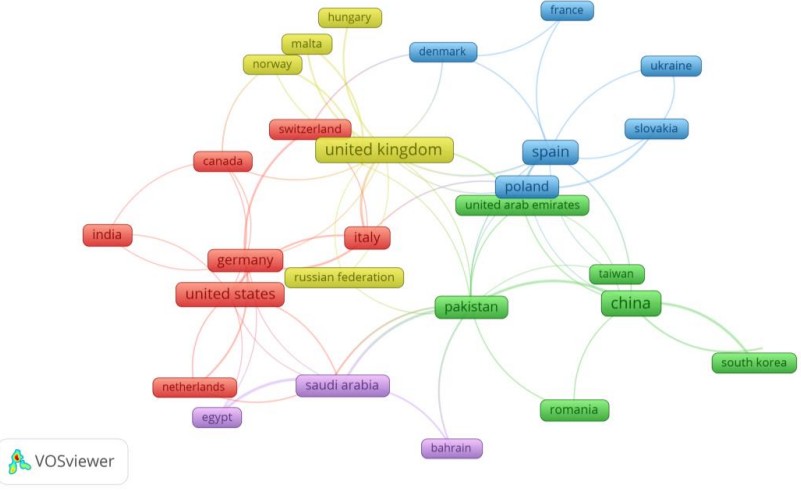

**Figure 9.** Scientific map of co-authorship analysis by countries.

### 3.3.3. Bibliographic Coupling

To determine the scope of scientific research dedicated to the integration of digital technologies in the activities related to CSR at enterprises, another bibliographic analysis tool will be applied—the so-called bibliographic coupling. When two articles cite a third article, then there is bibliographic coupling among them. In its essence, this type of analysis researches the shared references or citations among articles to identify the linked publications. By identifying publications that share common references, we can understand the degree of overlap of the literature and the mutual coupling of the research within a single area.

The VOSviewer tool has the capacity to analyze the bibliographic coupling of documents, sources, authors, organizations, and countries. In the research, we have applied only bibliographic coupling by documents.

The idea is to identify the current state of the researched problems in the 129 articles in the sample by reporting the use of identical scientific literature by their authors. In VOSviewer, we select only those 40 that are cited in Scopus at least 10 times. This will reveal links between every two articles, which will reflect the number of them simultaneously cited in both articles and other articles, which are identical to both of them. Twelve articles remain outside the selection because they do not have similar links to the other articles in the sample, which narrows the documents that are the object of bibliographic coupling to 28 items. They are distributed in clusters of different colors—red (13 items), green (8 items), and blue (7 items). Among all of them, there are 87 bibliographic couplings, which refer to citing a total of 198 literary sources.

The science map of the analysis is visualized with VOSviewer and is shown in Figure 10.

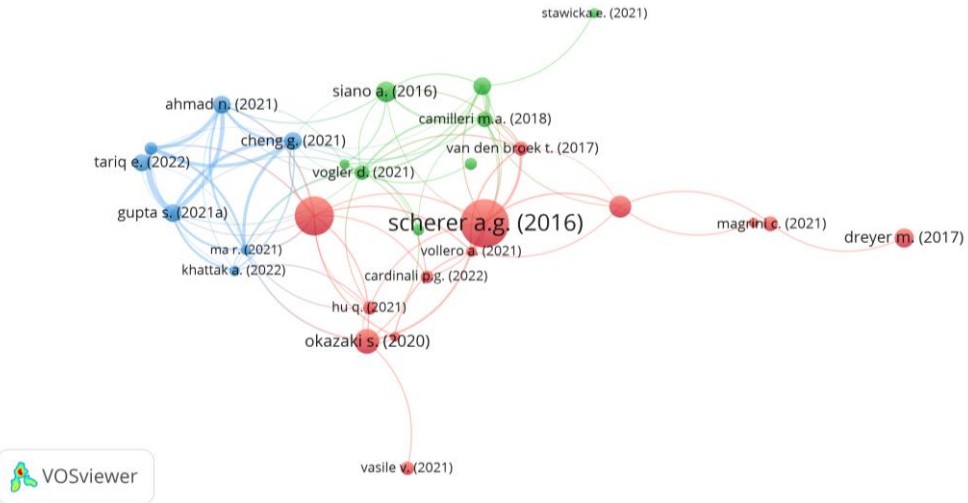

**Figure 10.** Scientific map of bibliographic coupling by documents. (Ahmad et al. 2021a; Camilleri 2019; Cardinali and Giovanni 2022; Cheng et al. 2021; Dreyer et al. 2017; Gupta et al. 2021; Hu et al. 2021; Khattak and Yousaf 2022; Ma et al. 2021; Magrini et al. 2021; Okazaki et al. 2020; Scherer et al. 2016; Siano et al. 2016; Stawicka 2021; Tariq et al. 2022; van den Broek et al. 2017; Vasile et al. 2021; Vogler and Eisenegger 2021; Vollero et al. 2021).

### 3.3.4. Co-Citation Analysis

The co-citation analysis can identify clusters of frequently co-cited publications, which can show newly emerged research fields or sub-fields. By studying the relationships among highly cited articles, we can understand the intellectual structure of the researched scientific area and potential directions for future research.

This type of analysis is based on a matrix of co-citation, which captures the frequency of co-citation among couples of articles. Every row and column in the matrix are a unique article, whereas the values of the cells show how many times two articles have been cited

together by other articles. As a result, a network of co-citation of articles is generated—a peculiar intellectual structure of the development of the studied scientific problems.

The VOSviewer tool has the capacity to analyze co-citations by cited references, cited sources, and cited authors. In this research, we have applied only a co-citation analysis by cited references.

VOSviewer requires setting a threshold, i.e., the maximum number of identical references (cited sources) cited by at least two articles in the sample. In the researched sample of 129 articles, the total number of references is 7830; 23 references correspond to three co-cited references, four of which are encountered four times in articles in the sample, whereas the remaining ones are cited three times. The co-cited sources were published between 1958 and 2021. The analysis of the co-citations outlines the relevant aspects of the problems of the digitalization of socially responsible business. The science map of the analysis prepared with VOSviewer is shown in Figure 11.

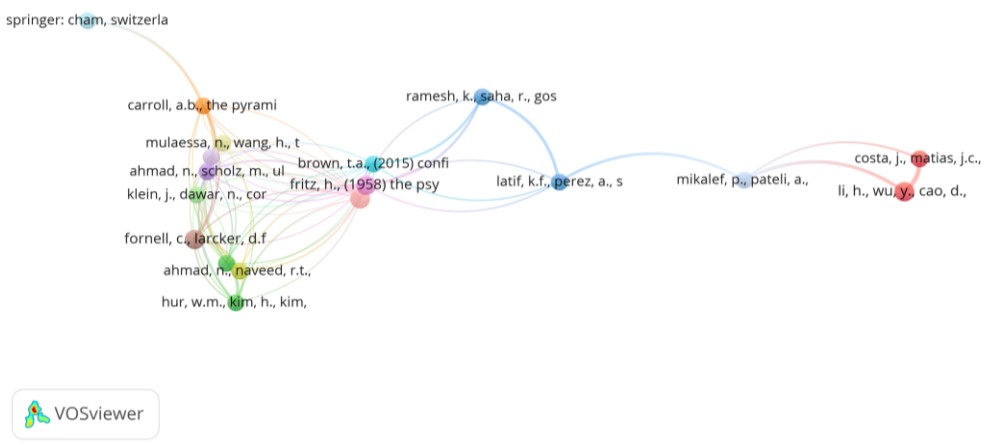

**Figure 11.** Scientific map of co-citation analysis. (Ahmad et al. 2021a, 2021b; Brown 2015; Carroll 1991; Costa and Matias 2020; Fornell and Larcker 1981; Fritz 1958; Hur et al. 2018; Klein and Dawar 2004; Latif et al. 2020; Li et al. 2021; Mikalef and Pateli 2017; Mulaessa and Wang 2017; Ramesh et al. 2019).

3.3.5. Co-Occurrence Analysis

The co-occurrence analysis focuses on the frequency at which certain terms or key words appear together in scientific publications. By identifying terms that appear frequently we can outline the scope and links within the researched scientific problem.

Out of the total number of 1127 key words in all articles of the researched material of this paper, we have initially selected only 51 words. The selection is a result of the VOSviewer option of selecting a minimum number of appearances of key words. We have set the software so that it selects a certain word or term at least three times as a key word in the articles in the sample. We have manually removed from the generated list of key words another 19 words because we have considered them irrelevant to the analysis (e.g., survey, article, human, male, female, etc.).

The remaining 32 words form four clusters with 160 links among them. The analysis continues by revealing the most frequently encountered key words. The product yields a list containing those words and the number of their appearances. The most frequently found words in publications are visualized by larger circles and are in the central part of the graphic.

The visualization shows the network, which is built by the key words in the researched area. The science map of the analysis is prepared with VOSviewer and is shown in Figure 12.

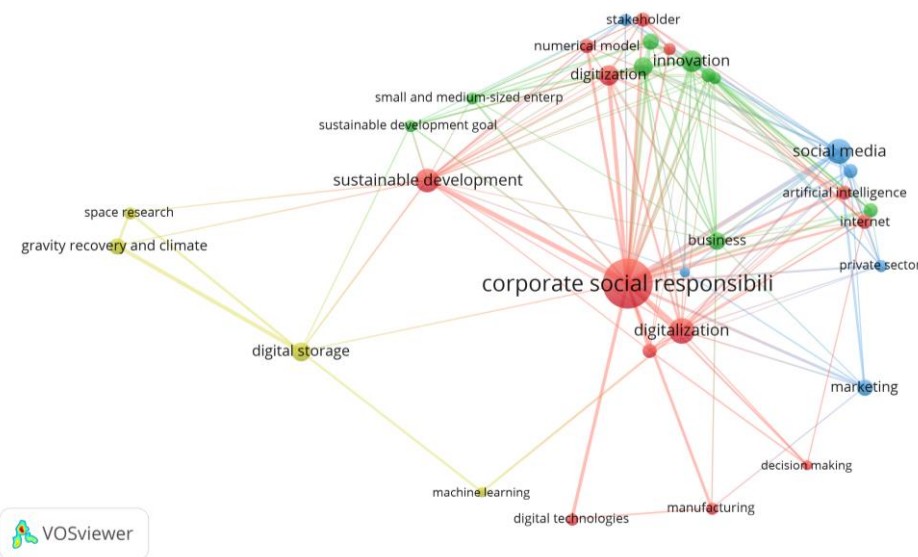

**Figure 12.** Scientific map of co-occurrence analysis.

## 4. Discussion

This research, regarding the posed research questions, has yielded results, which allow the following assertions to be made:

First, RQ1 refers to revealing the most influential publications, authors, and institutions, which research problems of integrating digital technologies in CSR activities. The analysis has shown that the article by A. G. Scherer, A. Rasche, G. Palazzo, and A. Spicer "Managing for Political Corporate Social Responsibility: New Challenges and Directions for PCSR 2.0." from 2016 happens to be the most influential publication with the most citations in Scopus—241. Its co-authors are affiliated with various institutions, namely The University of Zurich and The University of Lausanne (Switzerland), Copenhagen Business School (Denmark), and City University London (United Kingdom). Within the sample, the most cited works are those by S. Okazaki, K. Plangger, D. West, and H. D. Menéndez "Exploring digital corporate social responsibility communications on Twitter" from King's Business School, King's College London, and University College London (United Kingdom) and by Z. Orbik and V. Zozul'aková, "Corporate Social and Digital Responsibility" from The Silesian University of Technology (Poland) and Constantine the Philosopher University in Nitra (Slovakia), both from 2019. These results indicate the findings about the scope and trends in the field of CSR activities in our digital age, namely inclinations towards active involvement of politics and citizens in activities related to sustainability and social responsibility, imposing responsibility for the digital footprint on the stakeholders in CSR activities, and the active use of digital communication channels.

Second, RQ2 refers to the dynamics and the spread of research production in the researched area. The findings show that the research on CSR activities, which rely on digital technologies, develops exponentially. After 2020, we observe a sharp increase in the number of publications, which is explained by the occurrence of the global pandemic caused by the COVID-19 virus. The crisis has provoked businesses to seek solutions in realizing CSR activities, which guarantees the health and life of those who participate in them. It is this circumstance that is the reason for the increased interest in integrating digital technologies.

This assertion also reflects the results of the analysis of the territorial spread of scientific works. The ranking of countries with the most publications in the researched area is topped by China with the highest number of cited documents, followed by the UK and the USA. These countries can be said to be generators of innovative practices in the field of sustainable and socially responsible business and digitalization because they possess the necessary capacity of intellectual, economic, and social resources, scale, and geopolitical positions.

Third, RQ3 refers to outlining the scope, key topics, and trends for future research related to digital technologies and CSR. To outline the responses, we have used bibliographic coupling, co-citation analysis, and an analysis of key words and terms.

Bibliographic coupling generates three clusters—red, green, and blue. The problems researched in them also present interest. In the red cluster, the bibliographic links are by topics in the field of corporate social responsibility and ethics—communication with the stakeholders, sharing business and responsibility, loyalty, political aspects of CSR, and all this through the communication power of the media, the Internet of Things, green initiatives. The green cluster contains bibliographically linked scientific publications on corporate social responsibility—on the role of digital media and corporate websites in sharing CSR initiatives. The outlined topics of the articles in the blue cluster are related to CSR in various in their essence sectors—banks, hi-tech companies, small and medium enterprises, developing countries and economies, in which digital solutions are successfully implemented.

The co-citation analysis outlines the relevant aspects of the problems of digitalization in socially responsible businesses. They can be grouped as follows: (1) relevant problems of CSR—trust, transparency, loyalty between business and society; effects on civil behavior and opportunities for institutionalizing CSR; emotional engagement with the problems of CSR; brand image and brand attitude; (2) digital solutions for CSR—integrating social media; digital information processing; digital communication channels in a networked socially responsible society; and (3) sector application of CSR—the banking sector, environmental enterprises, the hotel industry, the Chinese economy.

The analysis of key words has revealed the most frequently used key words and terms in the articles in the sample in the outlined scientific area. These are CSR, sustainability, social media, and innovations. They outline the field, the "nucleus", and the state of the problems in scientific literature regarding the integration of digital technologies in CSR activities. Contrary to them, the most remote from the remaining key word and term notions shows the boundary values of the analysis, i.e., they are the least researched in the literature. These are, for instance, the world wide web, machine learning, Artificial intelligence, COVID-19, private sector, and space research. These types of key words mark the possible trends for future research, especially if they are combined among each other.

Fourth, RQ4 refers to looking for co-references of CSR and digital technology. The assumption that we were making in the beginning confirmed that the co-referencing of both categories in the key words implies that there is an integration of digital technology into the CSR activities.

As a result of the research, we can outline, as relevant sectors of further scientific research, the following: banks (Rangel-Pérez et al. 2023; Cheng et al. 2021), SMEs (Camilleri 2019; Stawicka 2021), family firms (Patuelli et al. 2022), human capital (Malynovska et al. 2022); innovative channels for communication with stakeholders, respectively, Twitter and others social media (Okazaki et al. 2020; Topor et al. 2022). At the same time, we can differentiate the following ethical aspects: customer trust, customer loyalty (Iglesias et al. 2020; Suebsaiaun and Pimolsathean 2018); new skills and competences (Abina et al. 2022).

Among the issues that remain outside the scope of this research is social network analysis as a way to more deeply understand the output of our research in the field of the development of collaboration network relationships among authors, organizations, and countries/regions. For this reason, the authors will deepen their research in this field in the future.

The limitation of the research is that there is a certain amount of subjectivity involved in selecting the sample for each statistic.

Research findings indicate that digital technologies strongly support CSR activities in modern corporate socially responsible business.

On the basis of the performed analysis, it can be concluded that the intersection between digital technologies and CSR is corporate digital responsibility, which can be visualized as follows (cf Figure 13):

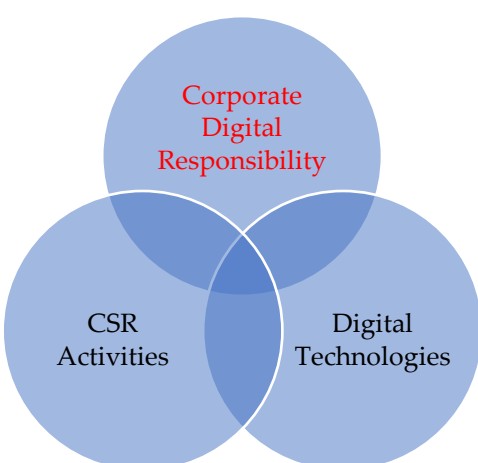

**Figure 13.** New business model for integrating digital technologies in CSR activities.

Corporate digital responsibility is the commitment of companies to manage their digital resources and technologies with responsibility and transparency in the context of the ethical, social, and legal issues associated with their use. This includes responsible conduct in data collection and processing, ensuring cybersecurity, and protecting the rights and freedoms of users (European Commission 2020). Corporate digital responsibility refers to the ethical and responsible behavior of companies concerning the use of digital technologies, data, and information systems. This includes protecting users' personal data, fair information processing, and avoiding digital activities that may pose threats to customers and society.

## 5. Conclusions

As a result of this research, we can outline two significant areas of possible future scientific and practical research:

First, specifying the key research areas and trends in integrating digital technologies in CSR activities, including the following:

- Digitalization of accounting and communication;
- Application of digital technologies in various aspects of the modern business;
- Innovations regarding digital solutions in the field of CSR (it is more frequently that companies use artificial intelligence, blockchain, the Internet of Things (IoT), virtual reality, and other technologies to improve the efficiency and impact of CSR activities);
- Knowledge sharing, collaboration, good practices (via online platforms, mobile applications, and networks among the stakeholders);
- Enhancing the quality of managerial decisions through more efficient monitoring, data analysis, and assessments of the efficiency and impact of CSR activities;
- Ethical and legal aspects of integrating digital technologies in CSR (problems of personal data protection, cybersecurity, and the responsible use of artificial intelligence);
- Strategies and the management of digital technologies in CSR (development of strategies and managerial models for integrating digital technologies in CSR);
- Social aspects of the process of integrating digital technologies in CSR (research on the social effects—the impact of competences, workplaces, education, access to technologies, digital inequality, and vulnerable groups).

Second, analysis of the most frequently used digital technologies in CSR activities and respectively their impact on organizations and the society, including the following:

- Social media and online communications (for popularizing CSR and communication with the stakeholders);
- Internet platforms for donations and volunteering (for engaging more employees and outsiders and management of the collected donations);

- Data analysis (for an assessment of the impact of CSR activities—data collection and analysis regarding the environmental footprint, social indicators, and ethical practices);
- Interactive educational technologies and resources (for encouraging education and the engagement of employees and other stakeholders—courses, video conferences, online events, and learning materials);
- Technologies for accounting and analysis (blockchain, digital tracking systems, and accounting, which guarantees transparency and lowers the risk of misuse).

A bibliometric analysis and systematic literature review outline the scope of the scientific and research interest, on the one hand, and the practical and applied interest, on the other hand, in the field of CSR. In addition, they form the basis of bringing forward the new trends for future publications, which shall upgrade and enrich the theory and practice.

**Author Contributions:** Conceptualization, A.A., G.C. and R.K.-H.; Data curation, A.A.; Formal analysis, A.A. and G.C.; Funding acquisition, G.C.; Investigation, A.A., G.C. and R.K.-H.; Methodology, A.A., G.C. and R.K.-H.; Project administration, G.C.; Resources, A.A. and R.K.-H.; Software, R.K.-H.; Supervision, R.K.-H.; Validation, A.A., G.C. and R.K.-H.; Visualization, R.K.-H.; Writing—original draft, R.K.-H.; Writing—review & editing, A.A. and G.C. All authors have read and agreed to the published version of the manuscript.

**Funding:** Institute for Scientific Research of Tsenov Academy of Economics, Svishtov, Bulgaria: 2023-03.

**Data Availability Statement:** The data presented in this study are available on request from the corresponding author.

**Conflicts of Interest:** The authors declare no conflict of interest.

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
