# Peer review of "Integration of Digital Technologies in Corporate Social Responsibility (CSR) Activities: A Systematic Literature Review and Bibliometric Analysis"

_jrfm, doi:10.3390/jrfm16080373_

Round 1

Reviewer 1 Report

I found the article to be very interesting but think that there is need for clarification. For instance, it was not until the very end of the paper that I finally understood what you were doing which is significanlty different that what you espoused. Of course, this assumes that I did, in fact, understand what yuo are trying to do. It seems that what you are really doing is looking for co-reference of CSR and digital technology. The assumption that you are making is that the co-referencing of these in the keywards implies that there is an integration of digital technology into the CSR activities. However, what this relationship is really needs to be spelled out. Further, how define digital technologies likewise needs to be clearly defined. 

At line 76 on page 3, I think that this is backwards. CSR activities are not able to implement digital technologies but rather digital technologies can be implemented to support CSR activities.

You make the assumption in your conclusions that the number of citations is an indication of the impact that the article has; maybe not. An article can be cited for either positive or negative reasons. A simple citation count says nothing about its influence only that people are discussing it.

I would like to see you define Corporate Digital Responsibility more. In my mind this has to do with things like data privacy and security, mining of individual information, AI, digital ethics, etc. I am not certain that this is what you mean.

You state that the growth curve of citations is exponential. You might want to look at this article:  Nikulin, C., Rubin de Celis, J., Cascini, G., Stegmaier, R., & Minutolo, M. (2011). Anticipation of technological change through patent analysis and logistic growth curve. As an example of how to consider the growth. It may be exponential at this point but it could be following a sigmoid curve.

You might want to look at some of the social network analysis papers as a way to understand the output of your research and / or the theoretical underpinnings.

The paper was / is very difficult to read. I would suggest some editing service to provide support. The sentence structure I found to be confusing.

Reviewer 2 Report

Abstract First sentence improving or improvement of

p. 1 significance of 'non-mandatory'? p.1, p. 19 Don't use 'etc.'

The purpose of the research is not clear.  

"The goal of this article is to identify the current state of research on the integration of digital technologies in CSR activities in business, as well as to prepare recommendations for further research." "In addition, companies use more frequently technologies to increase the efficiency and impact of their CSR activities."

"We defend the thesis that CSR activities are thoroughly theoretical and scientifically grounded and have the capacity to grow and successfully implement digital technologies – a practice which, without doubt, contributes to their accelerated development and implementation, with a horizon towards corporate social responsibility."

There is no connection of the technologies to actual CSR activities in business.  There is no definition of CSR.  You said that CSR activities and thoroughly theoretical and scientifically grounded but you do not explain that.  CSR now includes social, environmental and governmental goals that are commected with companies' missions.

I think this paper should be revised to just focus on the research that studies the use of technologies in business and delete the CSR references.

It would be helpful to have an editor read this and make edits to sentends tha are not clear and paragraphs that do not develop a specifice topic well.

Reviewer 3 Report

I think this study is valuable in that it systematically shows the research trend of the last decade on the use of digital technology in CSR research by selecting various methodologies that can exclude subjectivity. It has a certain value as a review paper on a new topic.

In addition, many review studies arbitrarily select and present important papers, but this study is impressive because it shows the list of important articles in Table 5 through an objective methodology.

However, although the methodology presentation was very detailed, I could not get a sufficient understanding of how the future research topics presented from line 543 to the end of the paper were derived. It's almost as if they were written down separately instead of being presented in the study.

In addition, the researchers devoted the text to too many different statistical attempts that not including some of them in the paper would not undermine the value of the study, resulting in the discussion and conclusions not being sufficiently supported by the body of the paper.

In general, I think the review paper could be more useful to readers by reducing the methodological and statistical presentations and focusing more on the descriptive part of the literature, which gives a better idea of what topics have been studied and what previous papers have done.

However, I think the paper is valuable in its current form, as it provides a good overview of the state-of-the-art in a particular area of research, and there are detailed suggestions for future research topics, although some of the derivation process is omitted.

Below are a few minor suggestions.

-       The paragraph starting at line 75 seems to be at the end of the introduction, where the conclusions or implications of the study would normally be presented. However, "We defend the thesis" does not seem a common phrase in research papers; therefore, I would like to ask you to reconsider this wording.

-       The methodology chosen sometimes involves subjective judgment (e.g., line 312), and most methodologies arbitrarily set thresholds and only count above them. At the end of the paper, I think it is necessary to discuss the limitation that there is a certain amount of subjectivity involved in selecting the sample for each statistic.

-       Sometimes the citation method seems to be inconsistent (e.g. line 36).

-       I believe there is a standardized way of citing references used by MDPI, so the citations in this study may need to be revised accordingly.

No specific problems were found.

Reviewer 4 Report

Thank you for giving me the opportunity to explore such interesting research.

The authors reflected their research in an article that meets the requirements of highly-reviewed journals. The strengths of the study are its relevance and the logical presentation of materials. The methodology is described in sufficient detail, the results are also quite logical, filled with visual representations of the study; the set research hypotheses and the presented logical framework for the paper always attract more attention and simplify the perception of the study.

However, I would like to make such suggestions for improvement:

1)      Figure 3. Articles published by country/region* - please, add the comment to the picture, mentioning, for example, TOP-3 countries;

2)      All the articles mentioned in Table 5 should be in references.

3)      there is no gap in the analysis of the literature; still concerning “corporate social responsibility”, it can be useful Dudek M. Methodology for assessment of inclusive social responsibility of the energy industry enterprises; human capital maybe this one will be useful Malynovska Y. Enhancing the Activity of Employees of the Communication Department of an Energy Sector Company.

4)      The Figures 2-12 are small and the the reader can hardly understand what is written there. Make them bigger.

5)      mention article’s limitation.

Round 2

Reviewer 1 Report

I am satissfied with the editing.

Author Response

Dear Mr/Mrs Reviewer,

We are very pleased with our hard work together. Your recommendations were very useful for us to make our research better. Thank you very much for your strongly support.

Your sincerely,

the Authors

Reviewer 2 Report

I don't find a version that identifies what changes have been made.  The only blue font is for the references.  I need a version that shows me in blue what changes have been made.

I don't find a version that identifies what changes have been made.  The only blue font is for the references. I need a version that shows me in blue what changes have been made.

Author Response

Response to Reviewer 2 Comments

Round 2

I don't find a version that identifies what changes have been made.  The only blue font is for the references.  I need a version that shows me in blue what changes have been made.

Please, see the attached file – this is the version with all identifications about changes we have made the previous time.

Point 1: Abstract First sentence improving or improvement of; p. 1 significance of 'non-mandatory'? p.1, p. 19 Don't use 'etc.'

Response 1: Thank you for the constructive comments and recommendations, which have improved the quality of our article. The authors improved the first sentence and the authors are careful with the use of the word "non-mandatory". The authors avoided the use of “etc.” on p. 1, p. 19 and others throughout the article.

Point 2: The purpose of the research is not clear. 

Response 2: The authors precisely defined the purpose of the research.

Point 3: There is no connection of the technologies to actual CSR activities in business.  There is no definition of CSR.  You said that CSR activities and thoroughly theoretical and scientifically grounded but you do not explain that.  CSR now includes social, environmental and governmental goals that are commected with companies' missions.

Response 3: Thank you for the constructive comments. We provided definitions for corporate social responsibility from various authors, substantiating that CSR activities are theoretically and scientifically grounded. Also, we added that CSR encompasses/ includes social, environmental, and governmental goals that are connected/ related to the companies' missions.

Point 4: I think this paper should be revised to just focus on the research that studies the use of technologies in business and delete the CSR references.

Response 4: Thank you for the constructive comment. The authors will focus their future research on the use of technology in business.

Point 5: It would be helpful to have an editor read this and make edits to sentends tha are not clear and paragraphs that do not develop a specifice topic well.

Response 5: The article has undergone a second round of language editing for quality improvement in English. During the initial submission to MDPI in English, we used the services of an English language translator who works as a lectur-er in a prestigious Bulgarian university. We have choosen him because he has a doctoral degree in English Philology and very good reputation in Bulgaria. In response to your note, we have provided the article for a second review by the same lecturer. We apologize that we do not have the financial possibility to use another editing service, as our salaries are very low in our country, and we have already spent our savings on translation. If we had known earlier that the editing services offered would save us the trouble of understanding the English text, we would not have used a translator on our side, but would have directly taken advantage of editing service of MDPI. For our future articles, we will know how to proceed from the very beginning. At the moment, we do not have additional funds for translation. We present the revised version - the article has been under second editing service by the lecturer, PhD in English Philology. We truly appreciate your understanding in the name of science and our desire for scholarly work. We would be extremely grateful if you could make a compromise now. We acknowledge our mistake and will utilize proven editing services in the future.

Once again, thank you for your valuable feedback and guidance, which has contributed to the improvement of our work. We genuinely appreciate your assistance in enhancing the quality of our research.

Round 3

Reviewer 2 Report

New material has met the requirements for changes.

Line 42 “others—as a new” Delete dash

Line 47 should say “industry and education”

Line 52 Digital technologies change all economic and social activities and result in the creation of new activities, services, innovation, and business opportunities.  Rewrite this sentence.  ‘creation of innovation’ does not work.  Would “creation of innovative ideas, services and business opportunities” be better in terms of your meaning?

Line 72 “communicate and function” function in what?

Line 74 “Ethical and responsible handling of data”—should be Ethical and Responsible Handling of Data.  Use capital letters to match the other headings

Line 94 should be “reducing the digital divide”

Line 124 are thoroughly theoretically and scientifically grounded

312 is China, delete is topped by Darkest blue is China delete (the darkest blue parts of the map.  Numbers are sufficient

616 rewrite ‘the authors consider to deepen their research…’ Rewrite ‘the authors will deepen their research….'

some changes need to be made as indicated above

Author Response

Response to Reviewer 2 Comments

Round 3

Point 1: Line 42 “others—as a new” Delete dash

Response 1: Thank you for the constructive comments and recommendations, which have improved the quality of our article. The dash is deleted.

Point 2: Line 47 should say “industry and education”

Response 2: Thank you for the constructive comments and recommendations, which have improved the quality of our article. We said “industry and education”.

Point 3: Line 52 Digital technologies change all economic and social activities and result in the creation of new activities, services, innovation, and business opportunities.  Rewrite this sentence.  ‘creation of innovation’ does not work.  Would “creation of innovative ideas, services and business opportunities” be better in terms of your meaning?

Response 3: Thank you for the constructive comments and recommendations, which have improved the quality of our article. We rewrote the sentence and included “creation of innovative ideas, services and business opportunities”.

Point 4: Line 72 “communicate and function” function in what?

Response 4: Thank you for the constructive comments and recommendations, which have improved the quality of our article. We rewrote the sentence and included “to function effectively and  efficiently”.

Point 5: Line 74 “Ethical and responsible handling of data”—should be Ethical and Responsible Handling of Data.  Use capital letters to match the other headings

Response 5: Thank you for the constructive comments and recommendations, which have improved the quality of our article. We used capital letters – “Ethical and Responsible Handling of Data”.

Point 6: Line 94 should be “reducing the digital divide”

Response 6: Thank you for the constructive comments and recommendations, which have improved the quality of our article. We added “the” – “reducing the digital divide”.

Point 7: Line 124 are thoroughly theoretically and scientifically grounded

Response 7: Thank you for the constructive comments and recommendations, which have improved the quality of our article. We added “ly” – “thoroughly theoretically and scientifically grounded”.

Point 8: 312 is China, delete is topped by Darkest blue is China delete (the darkest blue parts of the map.  Numbers are sufficient

Response 8: Thank you for the constructive comments and recommendations, which have improved the quality of our article. We deleted “topped by” and “(the darkest blue parts of the map)“. We agree with you that numbers are sufficient.

Point 9: 616 rewrite ‘the authors consider to deepen their research…’ Rewrite ‘the authors will deepen their research….'

Response 9: Thank you for the constructive comments and recommendations, which have improved the quality of our article. We deleted “consider to”. We rewrote the sentence – “the authors will deepen their research…”.

The article has undergone a third round of language editing for quality improvement in English.

Once again, thank you for your valuable feedback and guidance, which has contributed to the improvement of our work. We genuinely appreciate your assistance in enhancing the quality of our research.
